# Procedural Generation of Semantically Correct Levels in Video Games using Reward Shaping

**Luke Kerker**[1]**, Branden Ingram**[1]**, Pravesh Ranchod**[1]

2460117@students.wits.ac.za, branden.ingram@wits.ac.za, pravesh.ranchod@wits.ac.za

[1]**School of Computer Science and Applied Mathematics, University of the Witwatersrand**

## Abstract

The generation of video game levels traditionally relies on manual efforts from skilled professionals, resulting in significant expenses and time commitments. Procedural generation offers a solution by automating this process, reducing costs but potentially sacrificing designer control. The drawback of diminished control is that it has limited the widespread adoption of procedural generation due to concerns about the quality of the generated levels. Various approaches, including reinforcement learning and evolutionary algorithms, have been explored to address this limitation by improving how procedurally generated levels align with designer constraints. However, a key challenge remains in designing reward schemes or evaluation functions that accurately capture these constraints. To tackle this challenge, this paper proposes a system utilizing semantically appropriate reward shaping in a reinforcement learning setting for procedural content generation. By integrating an additional shaping function into the reward mechanism, this system generates diverse video game levels in the Zelda Gym environment that meet designers' specific requirements and constraints.

## 1 Introduction

Generating video game levels is an expensive, time-consuming, and labour-intensive endeavour, as many experienced and skilled professionals are required to manually generate high-quality content such as environments, object models, and substantially more Hendrikx et al. (2013). Procedural generation reduces the monetary, time, and labour costs of generating levels in video games by substituting the manual process with an automatic process, albeit at the expense of diminishing designer control over the automatically generated levels Van Der Linden et al. (2013); Togelius et al. (2011). Consequently, this limitation, along with others, inhibits the widespread commercial use of procedural generation for creating video game levels, as the reduced control designers have over the generated levels often results in undesirable and unusable low-quality levels Van Der Linden et al. (2013).

Procedural content generation algorithms are capable of producing playable and winnable levels that aim to comply with designers' overall goals through the use of reward schemes and evaluation functions Khalifa et al. (2020). The challenge confronting procedural content generation algorithms is designing reward schemes or evaluation functions that precisely capture the designers' unique constraints and requirements Khalifa et al. (2020).

To overcome this challenge, we propose a system capable of procedurally generating video game levels that are both diverse and constraint-compliant. This is accomplished through the implementation of reward shaping in reinforcement learning for procedural content generation, which incorporates an additional shaping function to enrich the reward mechanism. We demonstrate the controllability of the shaping function in the OpenAI Gym environment Brockman et al. (2016) against

standard procedural content generation through reinforcement learning. We successfully generated levels with a comparable degree of diversity to the traditional Procedural Content Generation via Reinforcement Learning baseline, while also achieving a higher level of constraint satisfaction than the baseline.

## 2 Background

### 2.1 Procedural Content Generation

Procedural content generation refers to the algorithmic creation of content Van Der Linden et al. (2013). The varying methods of procedural content generation are collected into the classes: Pseudo-Random Number Generators Perlin (1985), Generative Grammars Müller et al. (2006), Image Filtering Lefebvre & Neyret (2003), Spatial Algorithms Ebert et al. (2002), Modelling and Simulation of Complex Systems Hendrikx et al. (2013) and Artificial Intelligence Mateas & Stern (2005); Nareyek (2007); Skinner & Walmsley (2019). The methods in each class can be used to generate various different types of content Hendrikx et al. (2013) such as for stories Riedl & León (2009), music Plans & Morelli (2012), images Fadaeddini et al. (2018) and video game levels Nasir et al. (2024). Procedural level generation is regarded as one of the most widely applicable forms of procedural content generation as most video game genres can benefit from procedurally generated levels Liu et al. (2021); Hendrikx et al. (2013). More complex approaches such as machine learning techniques have also been applied to the problem of algorithmic content generation, such as supervised learning (PCGML) Summerville et al. (2018) and reinforcement learning (PCGRL) Khalifa et al. (2020). Employing reinforcement learning for procedural content generation is a relatively recent development Liu et al. (2021), with the major difficulty in its adoption being the existence of a training environment as well as the difficulty in designing the appropriate reward function.

### 2.2 Reinforcement Learning

Reinforcement learning, as described by Sutton et al. (1999), integrates concepts from psychology, engineering, and learning theories to develop algorithms that enable computers to learn optimal policies for maximizing long-term rewards Sutton et al. (1999). This framework centres on the interaction between agents and their environment, incorporating essential components such as the reward function, state-action space, and value function, which together guide agents in decision-making Sutton et al. (1999).

Sutton et al. (1999) states that a policy specifies how an agent behaves in a given state by defining the probability distribution over actions Sutton et al. (1999). The formulation for a deterministic policy is as follows:

$$\pi(s) = a$$

The symbol $\pi$ represents a policy that returns an action denoted by $a$ when in some state denoted by $s$. Mnih et al. (2013) define the optimal action-value function $Q^*(s, a)$ as the highest expected return attainable by following any policy, given a specific sequence $s$ and subsequently selecting an action $a$ Mnih et al. (2013). The optimal-value function is formulated as:

$$Q^*(s, a) = max_\pi \mathbb{E}[R_t | s_t = s, a_t = a, \pi]$$

Where the symbol $R_t$ denotes the discounted reward at time-step t. In many reinforcement learning algorithms, the core concept revolves around iteratively updating the action-value function based on the Bellman equation to converge towards the optimal action-value function Sutton (2018). However, directly estimating the action-value function for each sequence lacks practicality due to the absence of generalization. Hence, it's common practice to use a function approximator to estimate the action-value function instead Mnih et al. (2013).

Function approximators provide the ability to generalize across similar states and actions by learning compact representations, allowing for efficient estimation and storage Sutton (2018). Reinforcement

learning leverages function approximation in algorithms like Proximal Policy Optimization to learn policies within continuous action spaces by modelling them as parameterized distributions over potential actions Schulman et al. (2017). This use of function approximation marks a significant shift from traditional tabular methods, where every state-action pair requires explicit representation, limiting scalability Sutton & Barto (2018).

## 3 Related Work

### 3.1 Objective function design

According to Togelius et al. (2012), precisely measuring complex video game characteristics such as atmosphere and design cohesion to form accurate objective functions poses a significant challenge in procedural content generation (PCG). To address this, they propose a hybrid PCG method that integrates complementary techniques, leveraging each method's strengths and compensating for their weaknesses.

Similarly, Ferreira et al. (2014) propose a multi-population genetic algorithm approach, independently evolving user-selected game elements. Each element is assigned its own population, fitness function, and genetic operators, with optimal solutions combined to produce high-quality game levels. In both methods, multiple specialized fitness functions simultaneously evaluate distinct level characteristics, enabling detailed and accurate quality assessments.

Togelius et al. (2010) highlight challenges associated with optimizing composite fitness functions due to the difficulty in assigning appropriate weights to individual metrics without prior solution distribution analysis. As an alternative, they recommend multi-objective evolutionary algorithms, which explore Pareto fronts of non-dominated solutions—those that perform equally or better than all others across conflicting fitness functions.

### 3.2 Reinforcement Learning

Designing reward functions for reinforcement learning models has similar challenges to designing objective functions as Khalifa et al. (2020) state that reinforcement models can over-fit to generate a single optimal video game level due to poor reward function definition within the model. Khalifa et al. (2020) propose a parameter called the change percentage which defines what percentage of the level the agent is allowed to alter. The change percentage defines the degree of greediness that the agent can be when altering the video game level. Greedier agents make alterations that yield greater short-term rewards which prevents the agent from converging to a single optimal level design. Justesen et al. (2018) overcome the challenge of designing a reward function by using a deep reinforcement learning agent to play and grade the levels generated by the reinforcement learning agent. The level-generating agent generates levels based on difficulty, initially starting at difficulty 0. The player agent plays the level generated by the level-generating agent and if the agent completes the level then the player agent returns a positive reward to the level-generating agent, otherwise the player agent returns a negative reward to the level-generating agent. Our research aims to utilize the additional constraint methods proposed by Khalifa et al. (2020) and Justesen et al. (2018) to procedurally generate semantically correct levels through reinforcement learning.

### 3.3 Designer control

Linden et al. (2013) propose that effective control over procedural content generation ensures alignment with designers' envisioned qualities. They introduce a method for constraining procedurally generated levels using a "gameplay grammar" that reflects expected gameplay. By translating design requirements (such as character moves and relationships) into this grammar, designers generate player action graphs, guiding the procedural creation of game levels. Similarly, Mawhorter & Mateas (2010) propose an "occupancy-regulated extension" algorithm, which builds levels iteratively around designated "anchors" (potential player positions) to maintain designer-intended

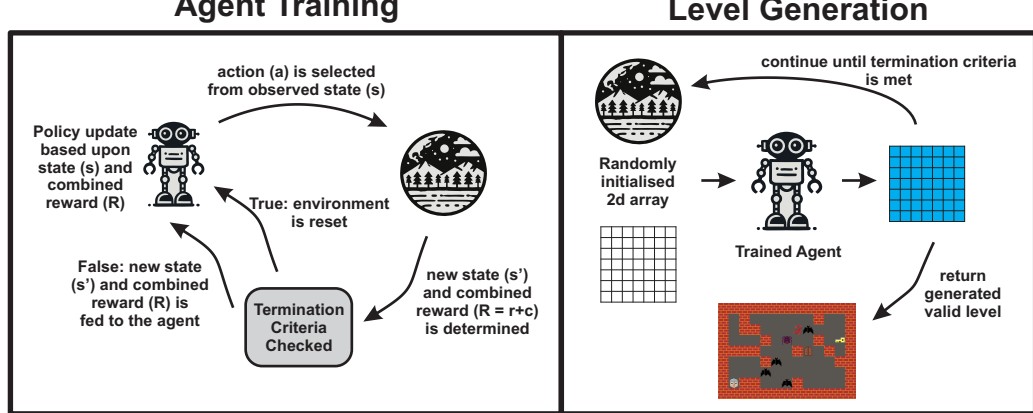

Figure 1: High-Level Pipeline of Shped PCGRL. Agent training involves a reinforcement learning algorithm that predicts actions from observed states, updates its policies based on a combined reward (R) incorporating both reward (r) and constraint (c) values, and continues until a termination condition is reached. Level generation describes how the trained constrained agent engages with the environment by sampling a random 2-dimensional array and performing actions over a set number of steps until a valid, playable level is created and returned as the final output.

gameplay. These grammar- and occupancy-based approaches give designers more direct control. However, a lack of understanding of fitness functions may reduce designers' control over generated content Van Der Linden et al. (2013).

## 4 Shaped Reinforcement Learning For Designer Constraint Compliance

We propose a shaped reinforcement learning approach that can be used to modify the environment's reward structure by integrating a shaping function. This modification has similarities with safety-constrained reinforcement learning and reward shaping Junges et al. (2016). The shaping function rewards the agent based on its compliance with predefined semantic constraints established by the designer. This is a similar goal to Earle et al. (2021), however, they primarily look to manipulate the path length based on designed goals. Our shaping function is integrated as follows:

$$R(s, a) = r(s, a) + c(s, a)$$

where $c$ denotes the shaping function that will evaluate the agent's action $a$ at state $s$ based on the designer's constraints, $r$ represents the environment's reward function, returning the reward for the agent's action $a$ at state $s$. Figure 1 presents an overview of the training and inference pipeline for a procedural content generation agent using shaped reinforcement learning. Unlike conventional methods, this approach uses a combined reward, integrating both rewards for functionality (level solvability) and constraints, to update the agent's policies. The pipeline for level generation begins by sampling a random two-dimensional array as a level, which the trained agent then modifies. After a set number of modifications, the level is evaluated for validity and playability; if these checks fail, a new sample is generated, otherwise, the level is returned to the user.

## 5 Experiments

### 5.1 Metrics

- Diversity: The Kullback-Leibler Divergence between the probability distributions of $n \times n$ tile patterns within a level is calculated. Specifically, for a given level, we generate a probability distribution encompassing all possible $n \times n$ patterns (where $n$ is a hyperparameter). The distance

between the two levels is

$$d(A, B) = \frac{1}{2}(KL(A||B) + KL(B||A))$$

Given $K$ levels, we compute the distance between each pair of levels and average these distances to obtain an overall diversity score for a particular diversity.

- Action Variance: Quantifies the variation in patterns chosen and positioned within the level Beukman et al. (2023). It involves determining the frequency of each pattern's selection across K levels and weighting it by the number of tiles comprising the pattern, considering that larger patterns occupy more space. The computation concludes with deriving the standard deviation of these weighted frequencies.

- Path Length: Assesses the level's complexity by calculating the shortest path from the player to the key objective, and then to the door objective, using Dijkstra's algorithm.

## 5.2 Model Training

The standard PCGRL baseline is trained using Proximal Policy Optimization implemented by Stable-Baselines 3. The implementation is directly adapted from Khalifa et al. (2020)'s GitHub repository. The baseline is trained for one billion steps as specified in the repository. The baseline reward function assigns a target range to each type of tile in the environment, defining ideal limits for each element's presence in a level. For instance, the target range for a player tile is set from 1 to 1, as a valid level requires exactly one player. In contrast, the range for enemy tiles might be from 4 to 8, as there is no strict requirement on the exact count of enemies. The reward function then assesses each tile type's current and previous counts relative to these target ranges, encouraging values to stay within or move closer to their respective ranges. The reward function encourages values to remain within or move closer to the target range, with specific rewards assigned based on the degree and direction of alignment or deviation.

### 5.2.1 Shaped Agents

All shaped agents are trained using the Proximal Policy Optimization reinforcement learning algorithm implemented by Stable-Baselines 3, with hyperparameters kept consistent with those of the baseline agent to maintain comparability. All agents are trained for one billion steps within the environment to ensure a fair evaluation. All shaped agents utilize the baseline reward function in its original form to facilitate the learning process for generating valid and playable levels. The shaping function is designed similarly to the reward function, employing target ranges for various constraint elements. Shaping functions can vary in complexity based on the requirements of the constraint itself. For simple constraints, a function may use a single target range to evaluate a specific attribute of the level. In contrast, more complex constraints may require multiple target ranges to assess and enforce multiple attributes of the level, ensuring effective and comprehensive implementation.

## 5.3 Environment

Our experiments are conducted within the OpenAI Gym environment Brockman et al. (2016). Specifically, we utilize the PCGRL OpenAI Gym Interface implementation by Khalifa et al. (2020), an adaptation of the OpenAI Gym framework that focuses on procedural content generation through reinforcement learning. The framework includes several video game environments where reinforcement learning techniques can be applied, including Binary, Dangerous Dave, MiniDungeons 1, Sokoban, Zelda, and Super Mario Bros. Our experiments are conducted within the Zelda environment, chosen specifically for its inclusion of enemies, which serves as a basis for certain constraint functions in our study. Our implementation can be accessed at (Redacted for anonymity).

### 5.3.1 Observation Space

The observation space in the Zelda environment includes the entire level as a seven by eleven two-dimensional array, with each element storing an integer from 0 to 7. Each integer corresponds to a specific tile type within the level. Table 1 presents the meaning of each integer within the two-dimensional matrix representing the level layout. Figure 2 presents an example of a two-dimensional array observation, while Figure 3 depicts the corresponding level representation derived from this array. Notably, Figure 3 reveals that the level is enclosed by a single layer of solid wall tiles, which are not included in Figure 2 as these tiles are not part of the observation space.

Table 1: Zelda Environment Tile Descriptions

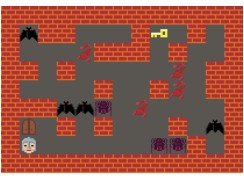

| 5 | 1 | 1 | 0 | 0 | 0 | 1 | 3 | 0 | 1 | 0 |
|---|---|---|---|---|---|---|---|---|---|---|
| 0 | 0 | 0 | 6 | 1 | 1 | 1 | 0 | 1 | 0 | 0 |
| 1 | 0 | 1 | 0 | 0 | 0 | 0 | 0 | 6 | 0 | 1 |
| 0 | 0 | 0 | 0 | 1 | 0 | 0 | 1 | 6 | 0 | 1 |
| 1 | 1 | 5 | 5 | 7 | 0 | 6 | 0 | 0 | 0 | 0 |
| 4 | 0 | 1 | 0 | 1 | 0 | 0 | 0 | 1 | 0 | 5 |
| 2 | 0 | 0 | 0 | 0 | 0 | 0 | 7 | 7 | 1 | 0 |

Figure 2: 2-Dimensional Array Level Representation

| Integer | Tile Description |
|---------|------------------|
| 0 | Empty (traversable) |
| 1 | Solid wall (non-traversable) |
| 2 | Player's spawning position (traversable) |
| 3 | Key, required to open the door (traversable) |
| 4 | Door, objective unlocked by key (traversable) |
| 5 | Bat, enemy (traversable) |
| 6 | Scorpion, enemy (traversable) |
| 7 | Spider, enemy (traversable) |

Figure 3: Level Representation Rendered from the 2-Dimensional Array in Figure 2

### 5.3.2 Action Space

In the Zelda environment, the action space is represented by an integer. The first and second digits correspond to the $x$ and $y$ coordinates, where the action will be applied. The $x$ coordinate can range from 0 to 11, and the $y$ coordinate from 0 to 7. The third digit specifies the type of tile to place at the designated $x$ and $y$ coordinate, limited to the values outlined in Table 1.

## 5.4 Semantically correct level generation

This study proposes categorizing semantic correctness into three distinct types: **quantity**, **locality**, and **structure**, each evaluated experimentally through constrained procedural content generation.

- For **quantity**-based correctness, an agent was trained to generate levels containing between eight and eighteen enemies, thus increasing enemy encounter frequency and overall level difficulty.

- To evaluate the combined quantity and **locality**-based correctness, an agent was shaped to position a specified number of enemies within four tiles of key objectives (keys and doors). Enemy proximity to objectives was measured using Dijkstra's algorithm, aiming to enhance difficulty by clustering enemies near critical objectives.

- For **structure**-based correctness, an agent was constrained to maximize solid wall tiles while ensuring levels consisted of a single contiguous traversable region. This constraint promotes narrow corridors and restricted movement areas, potentially heightening difficulty and encouraging diverse player strategies due to limited traversal options.

## 6 Results and Discussion

In each experiment, we generate one hundred levels per agent and repeat the experiment five times. For each set of one hundred levels, we calculate the action variance, level diversity, and path length, then average these metrics across the five repetitions. The level with the best metrics across repetitions is stored for qualitative analysis. A per-level analysis of the following results can be seen in Figures 10, 11, 12 depicting the mean and variance scores for each of the evaluated metrics.

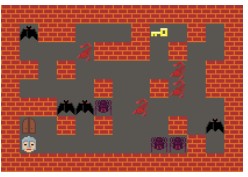

Figure 4: Baseline Agent - Level with greatest Enemy Count

Table 2: Metric Comparison Between Quantity Constrained and Baseline Agents

| Metric | Baseline Agent | Quantity Constrained Agent |
|---|---|---|
| Avg. Number of enemies | 4.442 | **7.362** |
| Avg. Action Variance | 1.143 | **1.851** |
| Level Diversity | 17.57 | **18.54** |
| Avg. Path Length | **22.88** | 14.428 |

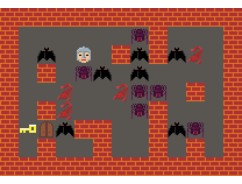

Figure 5: Quantity Constrained Agent - Level with greatest Enemy Count

## 6.1 Semantically correct quantity generation

Figures 4, 5 and 10 present the outcomes of the agent shaped to be semantically correct in regards to the quantity being evaluated against the baseline agent. Figure 4 shows the level with the highest enemy count generated by the baseline agent, while Figure 5 depicts the level with the most enemies generated by the shaped agent. Table 2 presents the evaluation results for both agents in relation to the metrics, with the "Avg. Number of enemies" serves as the constraint compliance metric. Figure 10 illustrates the performance of both the baseline and shaped agents, with each metric—constraint compliance, action variance, and path length—evaluated in its own sub-figure. These metrics are averaged over five independent runs, each encompassing the generation of one hundred levels.

The shaped agent is able to procedurally generate levels with greater enemy counts than the baseline agent, producing an average of three more enemies per level. By focusing on placing different enemy types, the shaped agent also achieved a marginally higher action variance. Additionally, the increased number of enemy tiles contributed to an average increase in level diversity. However, the shaped agent generated levels with shorter path lengths compared to the baseline, likely due to an imbalance in the reward-to-constraint function ratio, which prioritized enemy placement. This trade-off between the objectives of the reward and shaping functions is observed across all experiments.

Figures 4, 5 and 10 indicate that the shaped agent adhered to the semantic quantity constraint by generating levels with a significantly higher number of enemies. The shaped agent demonstrated enhanced performance across most metrics relative to the baseline, underscoring the effectiveness of shaping functions in achieving quantity-based semantic correctness.

## 6.2 Semantically correct quantity and locality generation

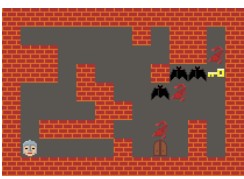

Figure 6: Baseline Agent - Level with the greatest number of Enemies Near Objectives

Table 3: Metric Comparison Between Locality-Shaped and Baseline Agents

| Metric | Baseline Agent | Locality Shaped Agent |
|---|---|---|
| Avg. Enemies near objective | 1.858 | **2.73** |
| Avg. Action Variance | 1.153 | **2.652** |
| Level Diversity | 17.57 | **18.371** |
| Avg. Path Length | **23.1** | 18.37 |

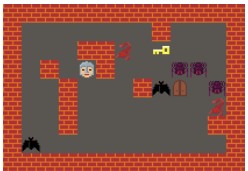

Figure 7: Locality Shaped Agent - Level with the greatest number of Enemies Near Objectives

Figures 6, 7, and 11 compare a shaped agent against a baseline on semantic constraints related to quantity and locality. The shaped agent consistently positioned, on average, one additional enemy near objectives compared to the baseline, resulting in increased action variance and level diversity. However, this focus also led to shorter player paths due to the shaping function prioritizing enemy proximity to key objectives. These results, clearly reflected in Table 3 and the figures, demonstrate the shaped agent's superior performance in constraint compliance and highlight the efficacy of shaping functions for semantically targeted procedural generation.

### 6.3 Semantically correct structure generation

Figures 8, 9, and 12 present an evaluation comparing the performance of a shaped agent against a baseline agent using a structure-based constraint. Levels generated by both agents were assessed based on the average number of solid wall tiles, action variance, and path length, as detailed in Table 4. The shaped agent consistently produced levels containing, on average, twenty-seven more solid wall tiles compared to the baseline, demonstrating strong adherence to the structural constraint. However, this constraint prioritization negatively impacted action variance and level diversity, resulting in fewer diverse actions and shorter path lengths due to an imbalanced reward-to-constraint ratio. Consequently, the shaped agent's generated levels featured narrow corridors and reduced traversable areas. These findings illustrate how stringent semantic constraints can adversely affect other performance metrics, emphasizing the need for careful balance in shaped reinforcement learning strategies for procedural content generation.

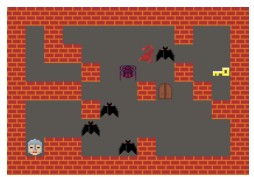

Figure 8: Baseline Agent - Level with greatest Solid Wall Count

Table 4: Metric Comparison Between Structure Shaped and Baseline Agents

| Metric | Baseline Agent | Structure Shaped Agent |
|---|---|---|
| Avg. Number of Wall tiles | 17.25 | **44.21** |
| Avg. Action Variance | **1.135** | 0.82 |
| Level Diversity | **17.57** | 15.70 |
| Avg. Path Length | **22.88** | 14.812 |

Figure 9: Structure Shaped Agent - Level with greatest Solid Wall Count

## 7 Conclusion

This study investigates the use of shaped reinforcement learning (RL) for procedural content generation (PCG), specifically targeting the automated creation of semantically correct video game levels aligned with designer-specified criteria. Traditional PCG methods, although economical, typically limit designer control, often compromising quality. The proposed approach integrates shaping functions within the RL reward mechanism, enabling the generation of levels that adhere closely to semantic dimensions—quantity, locality, and structure.

Results indicate that the shaped RL agents successfully generated levels meeting targeted design parameters, significantly outperforming baseline methods. These agents created levels featuring higher enemy density, strategically positioned enemies near critical objectives, and complex structural designs influencing player navigation and gameplay dynamics. However, there were observed trade-offs: emphasizing certain level features occasionally reduced path length and level diversity relative to the baseline, demonstrating a delicate balance between constraint adherence and overall level performance.

The findings emphasize the effectiveness of constraint-based RL methods in aligning procedural generation with intricate designer intentions. They also highlight the importance of future research directions, including fine-tuning the balance between rewards and constraints and exploring dynamic constraint adaptation during training.

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

# Supplementary Materials

*The following content was not necessarily subject to peer review.*

Figures 10, 11, and 12 illustrate the performance of the shaped models relative to the baseline on a per-level basis. The presented results reflect averages computed over 100 generated levels across five training runs, displaying both the mean and variance for each metric. These metrics; constraint compliance, action variance, and path length, align with the evaluation criteria previously discussed.

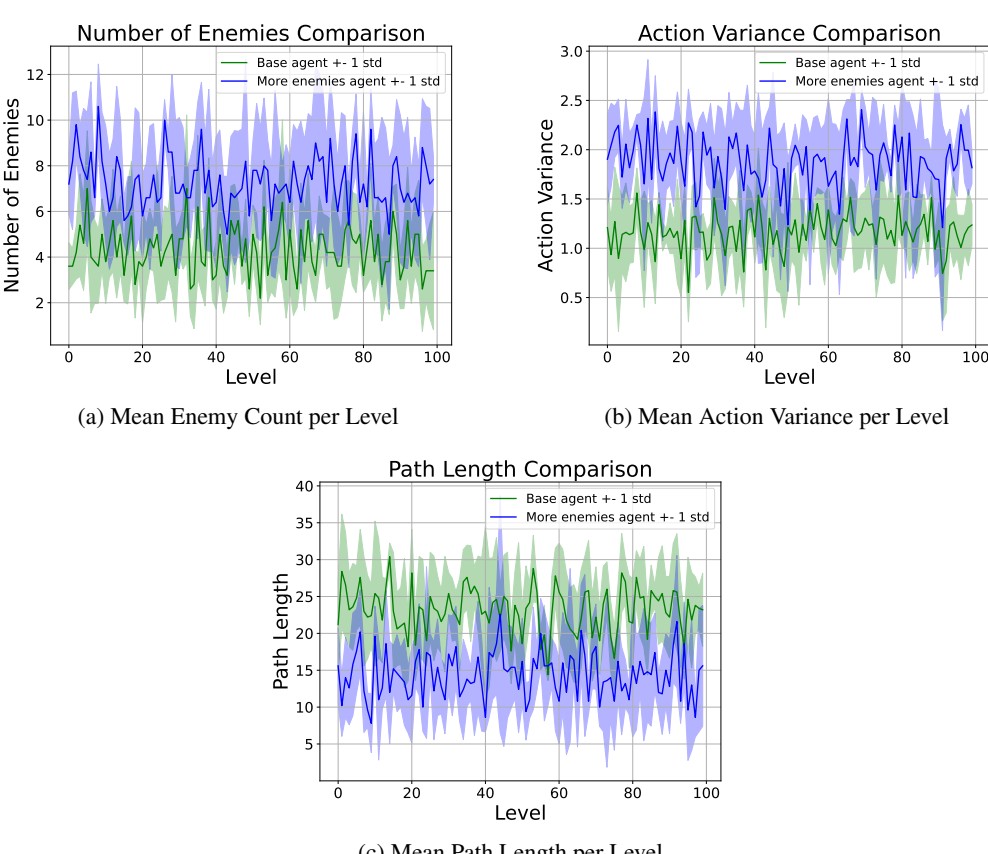

(a) Mean Enemy Count per Level

(b) Mean Action Variance per Level

(c) Mean Path Length per Level

Figure 10: Performance Metrics with ±1 Standard Deviation for Baseline and Quantity Shaped Agents, Averaged Over Five Runs Across 100 Levels. (a) Represents the constraint compliance metric, (b) displays the action variance, and (c) shows the path length.

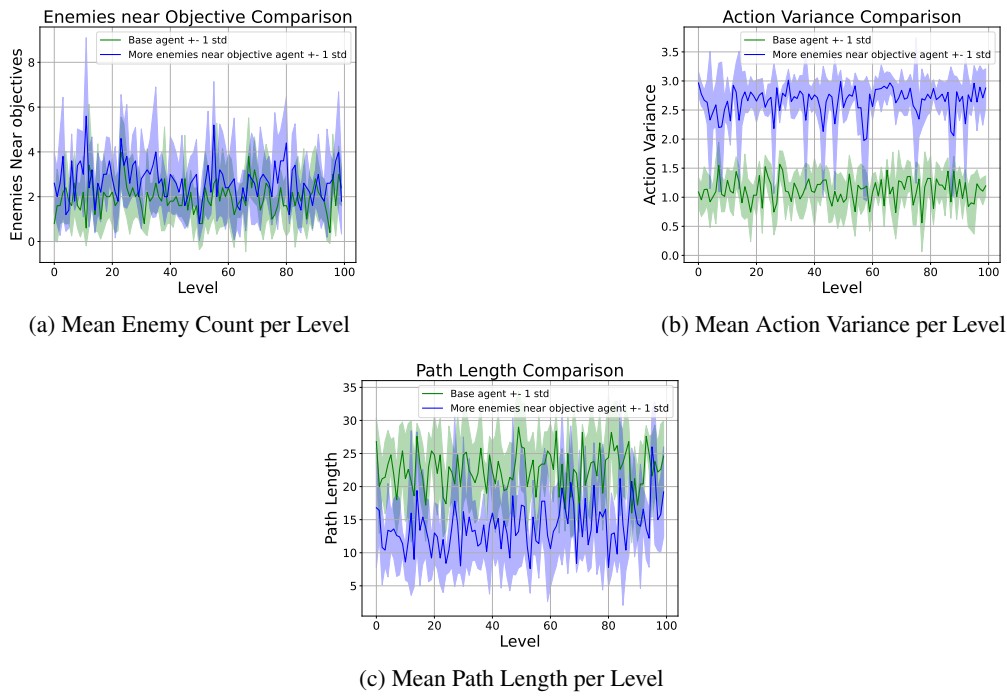

(a) Mean Enemy Count per Level

(b) Mean Action Variance per Level

(c) Mean Path Length per Level

Figure 11: Performance Metrics with ±1 Standard Deviation for Baseline and Locality Shaped Agents, Averaged Over Five Runs Across 100 Levels. (a) Represents the constraint compliance metric, (b) displays the action variance, and (c) shows the path length.

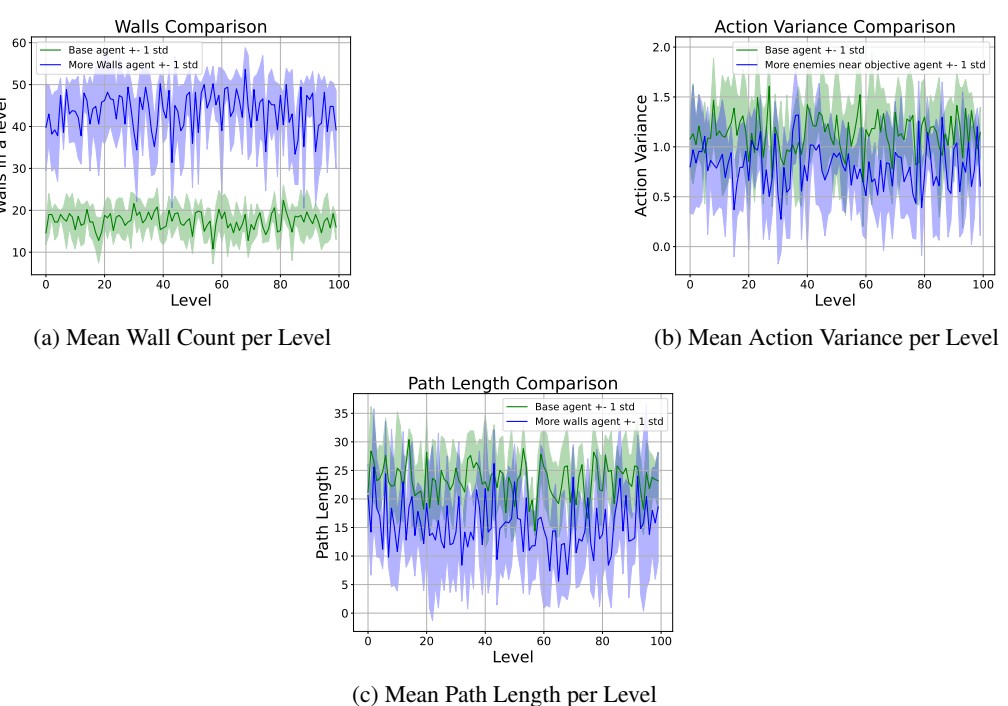

(a) Mean Wall Count per Level

(b) Mean Action Variance per Level

(c) Mean Path Length per Level

Figure 12: Performance Metrics with ±1 Standard Deviation for Baseline and Structure Shaped Agents, Averaged Over Five Runs Across 100 Levels. (a) Represents the constraint compliance metric, (b) displays the action variance, and (c) shows the path length.