# OpenReview forum: "Procedural Generation of Semantically Correct Levels in Video Games using Reward Shaping"
_rl-conference.cc/RLC/2025/Workshop/RLVG — RLVG Workshop - RLC 2025_

### Official Review · Reviewer_Jx9Z · 2025-06-15
**A good presentation of an alternative reward shaping framework for PCGRL. The method would be more easily understandable with some explicit examples of the reward component c(s,a).**

**Rating:** 4
**Confidence:** 4

**Summary:**

The authors present a reward shaping framework for Procedural Content Generation Reinforcement Learning (PCGRL) that enables higher design control and generates semantically correct levels compared to a baseline. The reward function constitute two terms $R(s,a) = r(s,a) + c(s,a)$. The reward term $r$ represents level solvability while $c$ measures semantic correctness. Within the scope of the work, semantically correct levels refers to levels which follow quantity, creating levels which contain 8-18 enemies; locality, enemies being within a distance to key objective (keys, doors) and structure, maximise solid wall tiles while also ensuring the existence of a single contiguous traversable path. The method was evaluated in the 2D tile-based Zelda environment.

**Strengths:**

The paper is well written and the method is presented and motivated succinctly. The background and related work facilitates a great overview of the PCGRL field. Using a similar framework as done in safe Reinforcement Learning and Constrained MDPs is an interesting avenue for further research for higher control of PCGRL methods.

The evaluation is extensive and the used metrics seem to be relevant for a PCGRL agent. The results highlights the strength of the method compared to baseline.

**Weaknesses:**

It is not clear from the paper how the authors' implementation of the semantic constraints looks like. It is never explicitly stated how $c(s,a)$ looks like each timestep which makes it harder to reproduce the work. As a reader, my guess would be that the agent receives a negative constant when the level have tiles outside of the specified ranges, and a positive constant inside the ranges.

**Best Paper Nomination:**

No

**Claims:**

The shaped RL agent can successfully generate levels that meet the targeted design parameters while also having higher diversity compared to baseline. While the method allows for a higher degree of level design control, there are still trade-offs regarding the ranges chosen for semantic constraints.

**Suggestions:**

While comparing the figures of the baseline and proposed method, I noticed that in these examples (Figures 4,5,6,7,8,9) the baseline agent usually placed the key passed the door away from the player compared to the shaped agent. Since the player has to go past the door for the key and then return to the door in the baseline cases, it can be argued that the levels are more "dynamic", since the required playstyle has more depth. However, it could be the case that the level with the highest semantic constraint value just does this naturally, but that the average placement of the shaped agent also showcase this behaviour of placing the key passed the door. It would be interesting to see if this is a consistent difference between the baseline and shaped agent.

It would be of interest to see how this framework works for larger environments with more types of tiles and a larger area. The 3D domain is also interesting, since semantic constraints can be harder to define and follow.

---

### Official Review · Reviewer_4Ra1 · 2025-06-17
**The contribution is reward shaping?**

**Rating:** 1
**Confidence:** 4

**Summary:**

The paper investigates reinforcement learning for procedural level generation, where challenges arise in diversity and quality of the generated levels, as well as regarding the alignment of the generated levels with the designer's preference.

To address the challenges, the authors propose to rely on shaped reward functions of the form
$$
R(s,a) = r(s,a) + c(s,a)
$$
that take the take the designer's semantic (?) preference into consideration. How to design such shaped reward functions is not stated clearly, it is only hinted at taking target ranges for specific metric into consideration. The proposed approach is then evaluated on a discrete 2D Zelda environment, where the quantity, locality, and structured in the generated levels is assessed. The results show that the proposed method takes the shaping function into consideration when generating levels.

**Strengths:**

1. The problem of procedural level generation under semantic constraints is challenging and worthwhile.
2. The related work section discusses relevant related works.

**Weaknesses:**

Unfortunately, there are many.

1. The paper uses sloppy notation and undefined symbols: In line 67, $s$ and $a$ are, I assume, elements of the state and action space, i.e. typically vectors that should be denoted in bold numbers according to standard notation.
2. Equations are not numbered.
3. Equation 2 (...) aka the one for the Q-function in line 71, is credited to Mnih et al. It should be credited to the original Q-learning paper. PPO is referenced but not cited at all.
4. The proposed reward function in line 135 clearly depends on the scale of the two terms. Why are there no coefficients? Unlike claimed by the authors, the equation for the shaped reward has nothing to do with safety-constrained RL, since it is not a constrained objective but rather a regular reward function.
5. The claim in line 140 that conventional methods don´t use combined reward functions is ridiculous, almost all but the most basic RL problems use reward functions that combine multiple terms.
6. The proposed method generates the complete 2D level. The scalability to more challenging levels is not discussed at all. As it stands, the method only provides evidence to be working for one toy example.
7. The diversity metric introduced on line 150 is questionable. What are the distributions A, B? What random variable are these distributions for?
8. It is unclear what observation and state spaces are described in 5.3.1 and 5.3.2. I assume these are for the level generator? Since the pipeline involves two agents (the generator and the one playing) this should be clarified.
9. The metrics under 5.4 are not well defined. For each of the three metrics, it is stated that how agents were trained for these metrics, but it is never explained what these metrics exactly measure and what their desired values are.
10. Related to the previous point, the interpretation of the results is hard. I can see that there are difference, but since the metrics are not  well defined and since their desired values are unknown, I can not tell which value is better.
11. The paper's main contribution seems to be reward shaping. Reward shaping is common practice for every RL practitioner and not a meaningful scientific contribution, especially since the paper does not even provide a clear heuristic for defining well-shaped reward functions for procedural level generation with RL.

**Best Paper Nomination:**

No

**Claims:**

The paper claims to provide a method that proceduraly generates levels which are "semantically correct", according to some designer imposed constraint, by means of reward shaping.

Unfortunately, the paper does not show how to do this, i.e. how to translate *semantic* constraints (e.g. "generate a level where enemies spawn in groups around structured terrain while ensuring valid paths through the level exist") into shaped reward functions. Instead, what this paper does is showing that RL policies will learn to select actions for which they are, *quantifiably and measurably*, rewarded, e.g. for generating levels with a certain number of elements in it. This is entirely expected. Challenges in reward shaping do indeed arise (as is mentioned in lines 97-101, 237 -240 and well agreed upon  in the RL community), however, the paper proposes no solution for addressing those challenges.

Thus, the claim that this paper provides an RL approach for procedural level generation under *semantic* constraints, does not hold.

**Suggestions:**

Maybe I misunderstood this paper, its narrative, and its contribution. That would indicate that the paper is extremely poorly written, which warrants rejection in itself.
If I understand it correctly, then the "contribution" is the proposition to shape reward functions to make RL behave a certain why. This is not a meaningful contribution to the RLVG community, which unfortunately also warrants rejection.

---

### Decision · Program_Chairs · 2025-06-19

**Decision:**

Accept

**Comment:**

This paper introduces a reward shaping framework for Procedural Content Generation via Reinforcement Learning (PCGRL). The reviewers appreciated the motivation behind the work and the thoroughness of the evaluation section. However, they also highlighted the lack of clarity and detail in the manuscript. We strongly encourage the authors to address these points, as well as the reviewers’ suggestions, in the camera-ready version.